# Improvement of Adsorption Capacity by Refined Encapsulating Method of Activated Carbon into the Hollow-Type Spherical Bacterial Cellulose Gels for Oral Absorbent

**DOI:** 10.3390/gels10110723

**Published:** 2024-11-08

**Authors:** Aya Hirai, Kaito Sato, Toru Hoshi, Takao Aoyagi

**Affiliations:** 1Department of Materials and Applied Chemistry, Graduate School of Science and Technology, Nihon University, 1-8-14, Kanda-Surugadai, Tokyo 101-8308, Japan; csay21002@g.nihon-u.ac.jp (A.H.); cska22016@g.nihon-u.ac.jp (K.S.); 2Department of Materials and Applied Chemistry, College of Science and Technology, Nihon University, 1-8-14, Kanda-Surugadai, Tokyo 101-8308, Japan; aoyagi.takao@nihon-u.ac.jp

**Keywords:** bacteria cellulose gel, activated carbon, seamless capsule, chronic kidney disease

## Abstract

To reduce the risk of adsorption of granular activated carbon (AC) in the gastrointestinal tract, we successfully prepared a hollow-type spherical bacterial cellulose gel encapsulated with AC (ACEG) and evaluated its pH tolerance and adsorption capacity. The bacterial cellulose gel membrane of ACEG features a three-dimensional mesh structure of cellulose fibers, allowing the selective permeation of substances based on their size. In this study, the preparation method of ACEGs was investigated, and the indole saturation adsorption capacity of the obtained gel was measured. We modified the gel culture nucleus gel from calcium alginate gel to agar gel, facilitating the encapsulation of previously challenging particles. The new preparation method used sodium hydroxide solution for sterilization and dissolution to remove the debris of *Komagataeibacter xylinus*, which was feared to remain in the bacterial cellulose membrane. This treatment was also confirmed to have no effect on the adsorption capacity of the AC powder. Therefore, this new preparation method is expected not only to improve the performance of ACEGs but also to be applied to a wide range of adsorbent-encapsulated hollow-type bacterial cellulose gels.

## 1. Introduction

In recent years, the number of patients with chronic kidney disease (CKD) has increased, and the number of patients requiring dialysis has also increased sharply [1]. In addition, CKD patients are at high cardiovascular risk [2]. There has been active discussion about measures to address these issues. Uremia is one of its symptoms and can cause CKD to worsen. Uremia is characterized by the retention of various solutes in the body that should be excreted by the kidneys. Among these solutes, those that adversely affect biological function are named uremic toxins [3], which are metabolic products produced from substances obtained through the diet and excreted in the urine by glomerular filtration or active transport by renal proximal epithelial cells [4]. However, if renal function is impaired, toxins accumulate in the body. These toxins are thought to be the cause of worsening uremia symptoms [5]. A variety of uremic toxins have been reported. Among them, indoxyl sulfate is a widely known representative substance [6] and has been shown to be toxic to various cells related to the pathogenesis of CKD, including osteoblasts [7], proximal tubules [8], vascular smooth muscle cells [9], and vascular endothelium [10]. Furthermore, indoxyl sulfate levels are frequently reported in papers dealing with uremic toxin concentrations in patients with kidney disease [3].

Indoxyl sulfate is known to be produced by the following process. Dietary tryptophan is converted to indoles by intestinal bacteria, and, after intestinal absorption, it is metabolized in the liver to produce indoxyl sulfate. In animal studies, indoxyl sulfate has been reported to reduce renal filtration function [11]. It was suggested that a vicious cycle occurs in which the indoxyl sulfate concentration in the blood rises further and renal function declines further because it is difficult to remove indoxyl sulfate from the blood due to the decline in renal function [12]. Therefore, suppressing the production of uremic toxins such as indoxyl sulfate is effective in reducing the progression of CKD and uremia. One of the ways to accomplish this goal is through the elimination of uremia using oral adsorbents.

Several substances are known as chemical adsorbents, among which activated carbon (AC) is widely used, and AC-based adsorbents are used to remove uremic toxins. Currently, AST-120 (KREMEZIN^®^), an oral adsorbent using AC, is used to delay dialysis initiation and improve uremic symptoms [13]. KREMEZIN^®^ is an AC-based adsorbent consisting mainly of 200–400 µm of spherical AC. However, the disadvantage is that it is not palatable and is difficult to swallow. KREMEZIN^®^ should be taken in 2 g doses three times a day. However, this dosage may decrease drug compliance due to discomfort during administration. Furthermore, oral ACs such as KREMEZIN^®^ adhering to the intestinal tract may result in risks of constipation, intestinal obstruction, and ulcers [14,15,16]. To avoid such risks, one option is to use more efficient adsorbents. Using small-particle-size ACs is one way to accomplish this because the smaller the particle size of the AC, the larger the specific surface area and pore volume [17]. However, ACs have small particle sizes and may be adsorbed in the intestinal tract, making them unsuitable for oral administration. In addition, patients with renal failure have limited fluid intake due to the burden on the kidneys. Therefore, it may be difficult to swallow powdered AC without discomfort. Therefore, there is a strong need for new dosage forms that improve palatability, are easy to consume, and minimize the amount required. Oral administration is said to be the most convenient and widely used method of drug administration [18]. Therefore, the development of a new oral adsorbent that is easy to swallow is a very important issue.

In this context, some reports have been published; for example, ager and alginate gel-type materials were used to encapsulate AC [19,20]. However, such soft materials have a risk of AC leakage in response to pH changes and peristaltic movements.

We have made great efforts to prepare hollow-type spherical bacterial cellulose (HSBC) gel and have succeeded in carrying out this action by devising innovative cultivation methods [21]. In order to improve the problem of dosing, we have succeeded in preparing AC-encapsulated hollow spherical bacterial cellulose gel (ACEG). In our previous report, we succeeded in producing ACEG containing AC particles with a diameter of 6 µm. Furthermore, the prepared ACEG was successfully evaluated for its adsorption capacity under various conditions and for its stability against pH for the uremic toxin precursor indoles and tryptophan [22]. In that experiment, sodium alginate was used to make a temporary AC nucleus. After completing encapsulation, the alginate was removed by immersion in a phosphate-buffer solution. As a result, the ACEG was stable even when stirred under acidic or basic conditions. Furthermore, indole and tryptophan were adsorbed even in the presence of inorganic ions. From these results, it is expected that the ACEG will pass through the digestive tract while adsorbing uremia-causing substances without leaking AC encapsulated in the digestive tract.

However, the amount of AC contained in our reported ACEGs is small compared to the amount of KREMEZIN^®^ in a single dose [22]. Therefore, we attempted to increase the amount of AC contained in one ACEG. It is suggested that the development of ACEGs containing more AC could reduce the number of gels required for administration. Therefore, it is hoped that promising new formulations will emerge that will reduce the daily dose and frequency of administration and minimize the associated risks.

In this study, we carried out the following investigations: First, we investigated the preparation method of ACEGs, aiming to increase the amount of adsorbent contained therein and expand its applications. Furthermore, we measured the amount of indole adsorbed by the obtained gels.

## 2. Results and Discussion

### 2.1. Comparison of Encapsulated AC Volume

In the previous preparation method, spherical alginate gel (Ca-Alg gel) was used. In this method, alginate reacts with calcium ions to form a cross-linked structure, producing a spherical gel containing AC. Therefore, it is necessary to drop a suspension of sodium alginate with AC into an aqueous calcium chloride solution. Aqueous sodium alginate solutions are highly viscous. The viscosity is further increased by the addition of AC. Therefore, increasing the AC content clogged the chips in the dripping procedure. This action made it difficult to increase the amount of AC encapsulated. Therefore, a preparation method using agar gel was used. There are two new methods. The main difference is the shape of the agar gel used. Method A uses agar gel cut into a cylindrical shape. Method B uses agar gel prepared into a spherical shape. The two methods are described in detail below. Agar is a kind of seaweed polysaccharide with gelatinizing ability and is mainly extracted from the Gracilariaceae and Gelidiaceae families [23]. Currently, agar is used in a wide range of applications, including food, pharmaceuticals, nutraceuticals, and biomedical applications [24]. When the agar solution is cooled, the agar molecules change from a linear to a double-helical shape, forming a stable three-dimensional network structure with increased hydrogen bonds. As the temperature drops further, the double helix coalesces to form a rigid gel [25,26,27]. The above mechanism allows agar to control its gel state and liquid state depending on the temperature. The agar solution in a heated state has a lower viscosity than the alginate solution. Therefore, it was suggested that ACEGs containing more AC than previous methods could be prepared. There are other advantages to changing from a Ca-Alg gel to an agar gel, such as the effect of ions. In the conventional method, a sodium alginate solution mixed with AC powder was dropped into a calcium chloride solution to form a Ca-Alg gel, which was then used as a nucleus for culture. After the BC gel membrane was produced, the Ca-Alg gel was dissolved and removed with a phosphate-buffer solution. In this method, the AC encapsulated in the ACEG comes into contact with various ions. This result suggests that the adsorption capacity of AC may be affected by ions. However, this is not necessary in the case of agar gel. As mentioned above, agar gels cause liquefaction and gelation depending on the temperature. Therefore, in the case of the method using agar gel, it is possible to prepare an AC-containing gel simply by changing the temperature. We theorized that the procedure for removing the internal gel after the production of the BC membrane could also be easily performed by using hot water. This action leads to fewer ions coming into contact with the AC. Thus, the effect of ions on the AC can be reduced. This action would increase the saturated adsorption of AC encapsulated in the ACEG and improve the performance of the ACEG as an adsorbent. For these reasons, agar gels were used for the new preparation method.

Photographs of the two gels obtained are shown in Figure 1. The AC content of each gel is shown in Table 1.

We will now describe Method A. We succeeded in preparing an AC-encapsulated BC gel with an improved volume per unit volume compared to previously reported gels. The maximum AC content in agar was 20 wt%. This result is because the viscosity of the agar solution increases with increasing AC content. As the amount of AC increased, the particles increasingly came into contact with each other. This result caused the water to lose its fluidity, become more viscous, and turn muddy. In this state, it could no longer be formed into a sheet and could not be hollowed out. As shown in Table 1, the amount of AC encapsulated per unit volume of the AC-encapsulated BC gel was successfully increased. Compared to the previous gel, which had a density of 35.3 [µg/mm^3^], we succeeded in preparing gel with a density of 65.1 [µg/mm^3^], which is approximately 1.84 times higher. However, when culturing was performed using the cylindrical agar gel as a nucleus, the BC gel membrane was sometimes torn at the edge of the cylinder during washing or in the process of removing the agar gel by heating and dissolving it. This result is because the agar gel that acts as the nucleus during culture has edges. When the gel was observed from the side after encapsulation with the BC gel, it was found that the thickness of the gel at the edges was thinner than that of the other parts (Figure 1b). In order to culture the BC membrane uniformly, it is necessary for the culture medium to adhere uniformly to the gel of the culture nucleus. Therefore, we decided to mold the agar gel nucleus into a spherical shape and culture it.

We will now describe Method B. To prepare spherical agar gels containing AC, an aqueous agar suspension was heated in silicone oil. When agar is dissolved by heating it in silicone oil, the surface energy causes the agar solution to form a sphere. We expected that cooling while maintaining the spherical shape would result in gelation. As a result, we succeeded in preparing the spherical AC-containing agar gel and culturing ACEG using the gel. As shown in Table 1, we succeeded in increasing the amount of AC included per unit volume of the ACEG. The amount of AC contained in the gel of Method A was 65.1 [µg/mm^3^], and the ACEG for Method B was 77.4 [µg/mm^3^]. The AC content of the ACEG was previously reported as 36.3 [µg/mm^3^]; thus, the AC content could be significantly increased by improving the preparation method. In order to further increase the content of AC, another method should be considered because, in the initial process to prepare the AC nucleus, the amount of AC that can be encapsulated is a maximum of 20 wt%, the same as in method A. If the concentration is increased beyond this value, contact between the cell suspension and the nucleus becomes incomplete, and we have confirmed that encapsulation using *Komagataeibacter xylinus* (*K. xylinus*, IFO13772) does not proceed well.

There are two possible factors for this result. The first factor is the change in the shape of the gel nucleus inside. As shown in Figure 2, when the gel nucleus inside is cylindrical, there is a thick part of the BC gel membrane (Figure 2, Method A). This part does not contain AC, so the amount of AC contained per volume of the ACEG decreases. On the other hand, when the gel nucleus is spherical, a thin and uniform BC gel membrane is produced. Therefore, the ACEG with a small volume and packed with AC is produced (Figure 2, Method B).

The second factor is the concentration of the AC-containing agar dispersion in the production method. As shown in Section 4.2.2, in the case of Method B, the AC-containing agar suspension is dropped and then heated to form a sphere. This action suggests that water evaporates more easily under high-temperature conditions. Therefore, the AC-containing agar gel prepared in Method B is considered to be more concentrated compared to Method A. Furthermore, by reducing the amount of the AC-containing agar suspension used in Method B, gels with smaller diameters could be prepared. This finding means that the gel size could be modulated by only changing the amount of dropped suspension. Therefore, Method B is considered to be a more practical preparation method.

In the previous preparation method, the Ca-Alg gel was used for the culture nuclei, but the Ca-Alg gel is cross-linked by ionic interactions. Therefore, when ions are held within the adsorbent itself, their inclusion in the gel is difficult. However, the new method uses agar gel, which is a thermoplastic gel. Even if the adsorbent holds ions, it is possible to prepare an encapsulating gel. Thus, it is expected that this technology will also be applied to adsorbents, which have traditionally been difficult to encapsulate. Therefore, this new preparation method will be useful not only for improving the performance of ACEG but also for the application of other adsorbent-encapsulated HSBC gels in a wide range of fields.

In this study, we focused on AC, a drug used to treat uremia, prepared the ACEG, and evaluated its indole adsorption capacity. However, the preparation method for the ACEG reported here is thought to be applicable to the encapsulation of other adsorbents. Ion exchange resins are an example of adsorbents used for oral medications. The development of ion exchange resins that are easier to swallow than conventional products is also underway [18]. Other interesting adsorbents to be encapsulated include zeolites. There are reports of oral administrations of zeolite to improve hyperglycemia, hyperlipidemia, and obesity in obese model mice [28]. In addition, a zeolite–polymer composite nanofiber mesh is being developed to remove uremic toxins for blood purification [29]. In addition, considering the hemocompatibility of cellulose materials in terms of coagulation and immune responses, the ACEG may also be applicable to remove uremic toxins from the blood. Therefore, the zeolite-encapsulated BC gel is expected to be useful in both oral medication and dialysis. Thus, the new ACEG preparation method reported here suggests that, by changing the nucleus gel from Ca-Alg gel to agar gel, it may be possible to greatly expand the applications of adsorbent-encapsulated BC gel.

### 2.2. Indole Adsorption Capacity Test

The relationship between each equilibrium concentration of ACEGs and the saturated adsorption amount is shown in Figure 3 for the measurement results of the gels of Method A and Method B.

Table 2 shows the saturated adsorption capacity calculated from Equation (3) for the previously reported gels, the gels of Method A, and the gels of Method B. Details of the calculations and equation transformations are described in Section 4.3.2. There are multiple methods for analyzing saturated adsorption using AC. In this study, as in previous reports, Langmuir’s adsorption isotherm equation was used in the analysis. As shown in Figure 3, the adsorption of indoles by encapsulated AC in both gels followed Langmuir’s adsorption isotherm. Under both conditions, *R*^2^ approached 1.0, suggesting that the adsorption of indole by ACEGs followed Langmuir’s adsorption isotherm. In this study, we prepared the ACEG and investigated its indole adsorption ability. Indole is a precursor of uremic toxins, and we thought that investigating its adsorption ability would enable us to investigate the usefulness of the ACEG as an oral adsorption drug. The ACEG produced by the new preparation method showed a higher saturated adsorption amount of indoles than previously reported. This result is thought to be due to the difference between the Ca-Alg gel and the agar gel. In the previously reported preparation method, the Ca-Alg gel is used as the nucleus, and after cultivation, the Ca-Alg gel is dissolved using a phosphate buffer. In this case, the AC may be affected by various ions. As an example, the AC is a popular adsorbent for removing Ca^2+^ from drinking water [30]. Therefore, it is suggested that AC encapsulated with ACEG may have lower adsorption capacity compared to the powdered state if it is affected by ions during the incubation stage. However, the new preparation method uses agar gel, which is a thermoplastic gel. This is believed to have reduced the influence of ions in contact with the AC surface during the preparation process, affecting the saturated adsorption amount of indole.

### 2.3. Effect of NaOHaq Washing on Adsorption Capacity

BC gel is produced by *K. xylinus*. When ACEGs were not treated with *K. xylinus* after collection, the *K. xylinus* produced BC gel during storage, causing ACEGs to stick to each other. In addition, when considering the practicality of ACEGs as an oral drug, the persistence of *K. xylinus* on the BC gel membrane is a problem that should be prevented. Therefore, the sterilization and lysis of *K. xylinus* were added to the preparation process. In biomedical applications, the sterilization of bacterial cellulose is required [31]. In this study, immersion in 1.0 wt% sodium hydroxide aqueous solution (NaOHaq) was used [32]. However, as mentioned above, there is concern that contact between AC and ions may affect the saturated adsorption capacity. Therefore, the effect of NaOHaq on the saturated adsorption of AC was investigated using powdered AC.

Figure 4 shows the relationship between the equilibrium concentration and the saturated adsorption amount for the two types of AC powders. Table 3 shows the saturated adsorption amount *Q_m_* calculated from Equation (3). As shown in Figure 4, the adsorption of indoles followed Langmuir’s adsorption isotherm for both AC powders. From the *Q_m_* in Table 3, it was found that immersion in NaOHaq did not affect the saturated adsorption of indoles on AC powder. Therefore, immersion in NaOHaq does not affect the inherent adsorption capacity of AC. Therefore, in the ACEG preparation procedure, the corpses of K. xylinus remaining on the BC gel membrane can be removed without affecting the amount of AC adsorbed.

## 3. Conclusions

ACEG, which uses agar as a culture nucleus, can encapsulate more AC than previous methods. The new preparation method includes sterilization and dissolution treatment using NaOHaq. This procedure removes the remains of *K. xylinus*: there was some concern that these would remain in the BC membrane. This procedure does not affect the adsorption capacity of normal AC powder. Therefore, the adsorption capacity of ACEG-wrapped AC is also expected to be unaffected. From the above, ACEG prepared using the new preparation method is not affected by ions, which was of concern with the previous preparation method. Therefore, an improvement in the saturated adsorption amount is expected.

These results indicate that the new preparation method makes it possible to prepare ACEG with higher performance. Therefore, this new ACEG preparation method could potentially contribute to the development of new formulations with improved performance compared to conventional methods. Furthermore, by changing from a Ca-Alg gel to an agar gel, which hardens when the temperature changes, it is expected that the method will be applicable to particles that were previously difficult to encapsulate.

Therefore, this new preparation method will be useful not only for improving the performance of ACEGs but also for the application of adsorbent-encapsulated HSBC gels in a wide range of fields.

## 4. Materials and Methods

### 4.1. Materials

The medium was prepared as follows: 45.0 g of D-glucose (Kanto Chemical Co., Inc., Chou-ku, Tokyo, Japan), 2.5 g of mannitol (Kanto Chemical Co., Inc.), 2.5 g of polypeptone (HIPOLYPEPTONETM, Nihon Pharmaceutical Co., Ltd., Chuo-ku, Tokyo, Japan), 2.5 g of Bacto^TM^ yeast extract (BD Biosciences, Franklin Lakes, NJ, USA), and 0.5 g of magnesium sulfate heptahydrate (MgSO_4_∙7H_2_O, Kanto Chemical Co., Inc.) were mixed in a concentration of 500 mL of Milli-Q water. AC (coconut-shell-activated charcoal, KD-PWSP, average diameter 6.0 µm, BET surface area 1.324 × 103 m^2^/g) was obtained from AS ONE Co. (Nishi-ku, Osaka, Japan). Silicone oils (KF-56A: 0.995 g/cm^3^, 15 mm/s^2^, ethanol-soluble oil) were purchased from Shin-Etsu Chemical Co., Ltd. (Chiyoda-ku, Tokyo, Japan). Other reagents were purchased from Kanto Chemical Co., Inc. and used as received. The aforementioned organic and inorganic chemicals were reagent grade, and the yeast extract was certified via analysis by the company.

### 4.2. Preparation of the HSBC Gel Encapsulating AC

In this study, we prepared ACEGs with the aim of increasing the amount of adsorbent encapsulated. To this end, we used two encapsulation methods. The first method involves culturing and encapsulating the BC membrane on the surface of a cylindrically cut agar gel. The second method involves culturing and encapsulating the BC membrane on the surface of agar gel molded into a spherical shape.

#### 4.2.1. Method A

First, agar gel was prepared as the nucleus for culture. The procedure is shown in Figure 5. AC powder was added to a 5 wt% agar solution. The solution was heated and stirred to dissolve the agar and then spread onto a Petri dish. The mixture was then pressed down from above, using a Petri dish with a 0.5 mm thick metal plate attached, and rapidly cooled. The obtained plate-shaped agar gel was hollowed out with a plastic needle (inner diameter 1.37 mm) to prepare a cylindrical agar gel containing AC. Next, agar gel was used as a nucleus for culture. The procedure is shown in Figure 6. The medium was sterilized by autoclaving, and *K. xylinus* was cultured at 30 °C for 3 days. The agar gel was soaked in a suspension of cultured cells, and *K. xylinus* was cultured at 30 °C for 1 day. Half the volume of silicone oil was poured into each well of a U-bottom 96-well plate (made of PTFE), and the spherical agar gel with the cell suspension remaining on the gel surface was immersed in it. While maintaining this condition, *K. xylinus* in the cell suspension was cultured at 30 °C for 1 week. On the first day of culture, 3 µL of the *K. xylinus* suspension was dropped onto the gel. On the fifth day of culture, the gel was turned upside down, and 3 µL of the *K. xylinus* suspension was dropped onto the gel, followed by agitation using a plate shaker (1000 rpm, 3 min). The addition of cell suspension was carried out on the 1st day and after 5 days of culturing. The suspension was the same one as the initial cell suspension soaked with the gels. After removing the BC gels biosynthesized at the surface of the culturing Petri dish, only the medium was used. After culturing the BC gel membrane, the gel was immersed in 1 wt% of NaOHaq to kill and remove the *K. xylinus*. Furthermore, Milli-Q water and the gel were placed in a covered Erlenmeyer flask and then left to stand in a small environmental tester at 95 °C for 24 h. This method removed the agar gel.

#### 4.2.2. Method B

The method for preparing spherical agar gel is shown in Figure 7, and the method for preparing ACEG using the gel is shown in Figure 8. Each well of a U-bottom 96-well plate (made of PTFE) was filled with silicone oil and placed on a metal tray. A 10 µL aliquot of 20 wt% AC-agar suspension (19 g Milli-Q water, 1 g agar, 5 g AC) containing the adsorbent suspended in it was dropped into each well. After heating in a small environmental tester at 120 °C for 30 min, the well plate was transferred to an ice bath and left to stand for 90 min. Using this method, spherical agar gels containing AC were prepared. The resulting agar gel was washed with Milli-Q water. The medium was sterilized by autoclaving, and the *K. xylinus* was cultured at 30 °C for 3 days. The spherical agar gel was immersed in a suspension of cultured cells, and the *K. xylinus* was cultured at 30 °C for 1 day. The spherical agar gel with the cell suspension remaining on its surface was immersed in each well of a U-bottom 96-well plate filled with silicone oil. While maintaining this condition, the *K. xylinus* in the cell suspension was cultured at 30 °C for 1 week. On the first day of culture, 4 µL of the *K. xylinus* suspension was added dropwise. On the fifth day of culture, the gel was turned upside down, and 4 µL of the *K. xylinus* suspension was dropped onto it. The addition of cell suspension was carried out on the 1st day and after 5 days of culturing. The suspension was the same one as the initial cell suspension soaked with the gels. After removing the BC gels biosynthesized at the surface of the culturing Petri dish, only the medium was used. After culturing the BC gel membrane, the *K. xylinus* were killed in the same manner as in Method A, the agar gel nucleus was removed, and the ACEG was obtained.

### 4.3. Comparison of Previously Reported Gels with Gels from Methods A and B

#### 4.3.1. AC Content Comparison

The amount of AC contained within the gel was measured using the following method. A total of 100 pieces of AC-containing agar gel were prepared using the same method as that used for the culture. After dissolving the gel and filtering the solution, the amount of adsorbent remaining on the filter paper was measured and used as the amount of AC contained within the ACEG.

#### 4.3.2. Indole Adsorption Test

To evaluate its performance as an adsorbent, the saturated adsorption amount of AC in the ACEG was evaluated. Indole, a precursor of uremic toxins, was used as the adsorption target substance.

Adsorption experiments were carried out using aqueous indole solutions at different concentrations. Measurements were performed at various concentrations until adsorption reached saturation. Adsorption isotherms were then plotted, and adsorption was deemed to have reached saturation when the line became flat. In the three conditions, the following indole concentrations were used: Method A, 5~50 µg/mL; Method B, 100~1000 µg/mL; and AC powder, 250~1000 µg/mL. Compared to Method B, Method A contained less AC per gel, which is thought to have led to saturation at a lower concentration range.

Before the test, the ACEG was immersed in Milli-Q water for 3 h in a small environmental test chamber at 70 °C. An amount of 5.0 mL of an indole solution and one ACEG were added to each container, and the mixture was stirred at 37 °C for 48 h using a plate shaker in a small environmental test machine. The concentrations of the solutions before and after adsorption were quantified using the following procedure. First, the absorbance at the maximum absorption wavelength (λ_max_ 270 nm) of aqueous indole solutions of known concentrations was measured using a UV–Vis spectrometer (V-630, V-730, JASCO Corporation, Tokyo, Japan). The relationship between absorbance and indole concentration was then plotted on a graph to create a calibration curve. The absorbance of the solutions before and after adsorption was measured, and the concentration was calculated using the calibration curve.

The equilibrium adsorption amount per unit mass of adsorbent at each concentration, *Q_e_* [mg/mg], was calculated using Equation (1), as follows:(1)$$Q_{e}=\frac{\left(C_{i}-C_{e}\right)V}{m}$$
where *C_i_* and *C_e_* [mg/mL] are the initial and equilibrium liquid phase concentrations of the uremic toxin precursors, respectively; *m* is the weight of the encapsulated AC [mg]; and *V* is the volume of the solution [mL].

Next, to evaluate the adsorption capacity, we applied Langmuir’s adsorption isotherm [33,34,35,36,37], and the saturated adsorption amount was calculated according to Equation (2), as follows:(2)$$Q_{e}=\frac{Q_{m}KC_{e}}{1+KC_{e}}$$
where *Q_m_* represents the practical limiting adsorption capacity (saturated adsorption amount) when the surface is completely covered with adsorbed molecules; *Q_e_* is the equilibrium adsorption amount [mg/mg] corresponding to complete monolayer coverage of the surface; *C_e_* is the adsorption equilibrium concentration [mg/mL]; and *K* is the Langmuir constant. Linearly rearranging Equation (2) produces the following equation (Equation (3)):(3)$$\frac{C_{e}}{Q_{e}}=\frac{C_{e}}{Q_{m}}+\frac{1}{KQ_{m}}$$

*Q_m_* was obtained from the slope of the plot of *C_e_*/*Q_e_* versus *C_e_*.

#### 4.3.3. NaOHaq Washing for Sterilization and Lysis of *K. xylinus*

There is concern about the effect of immersion in NaOHaq on the pore structure of AC. If the pore structure of AC is damaged, there is a risk of a decrease in adsorption capacity. The AC powder was prepared as purchased. The AC powder was soaked in 1 wt% of NaOHaq for 3 days, and, then, it was washed and dried. An amount of 30 mL of aqueous indole solution and 20 mg of AC powder were added to each vessel and stirred for 24 h using a plate shaker at 37 °C in a small environmental tester. The indole concentration in the solution was then measured using the same method as above, and the saturated adsorption amount was calculated.

## Figures and Tables

**Figure 3 gels-10-00723-f003:**
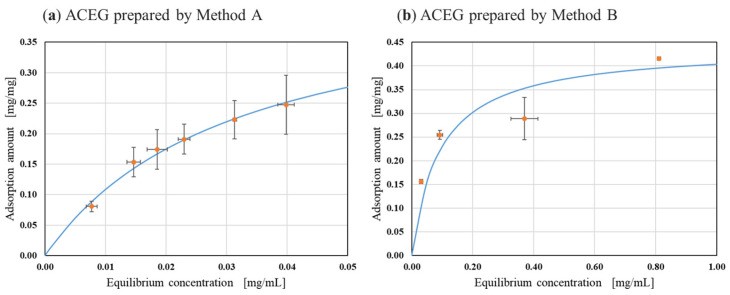
Adsorption isotherms of encapsulated AC. The orange plots are experimental values. The blue line is the curve fitted to the Langmuir equation using the parameters in Table 2 (n ≧ 3).

**Figure 4 gels-10-00723-f004:**
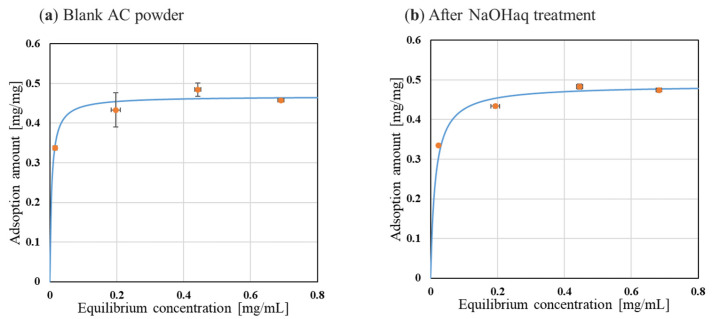
Adsorption isotherms of AC powder. The orange plots are experimental values. The blue line is the curve fitted to the Langmuir equation using the parameters in Table 3 (n = 3).

**Figure 1 gels-10-00723-f001:**
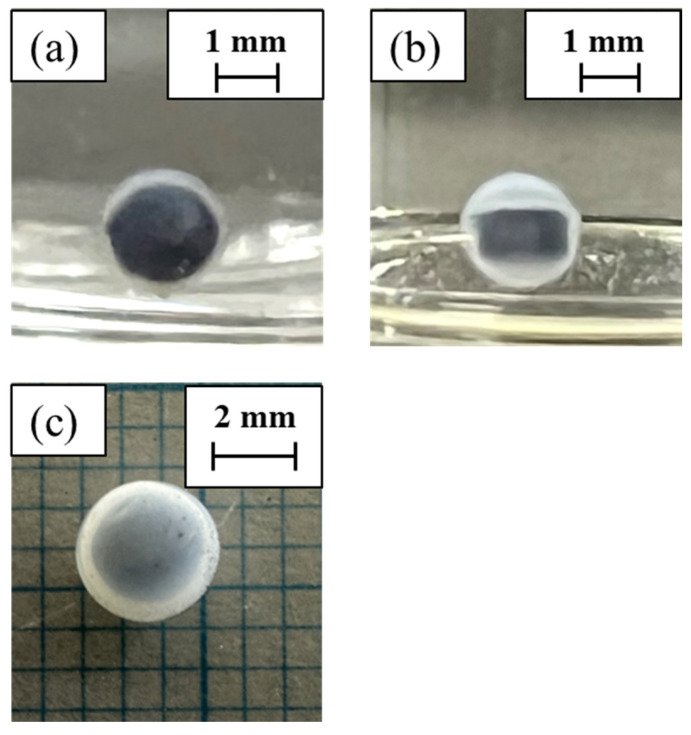
Photographs of (**a**) top view and (**b**) side view of ACEG prepared by method A. (**c**) ACEG prepared by Method B.

**Figure 2 gels-10-00723-f002:**
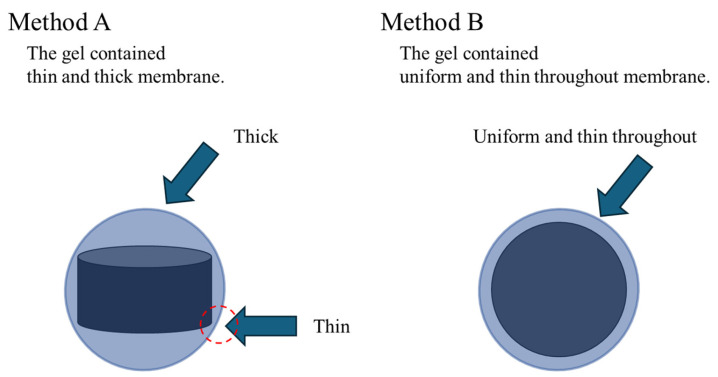
Schematic representation of ACEG depending on the shape of nucleus.

**Figure 5 gels-10-00723-f005:**
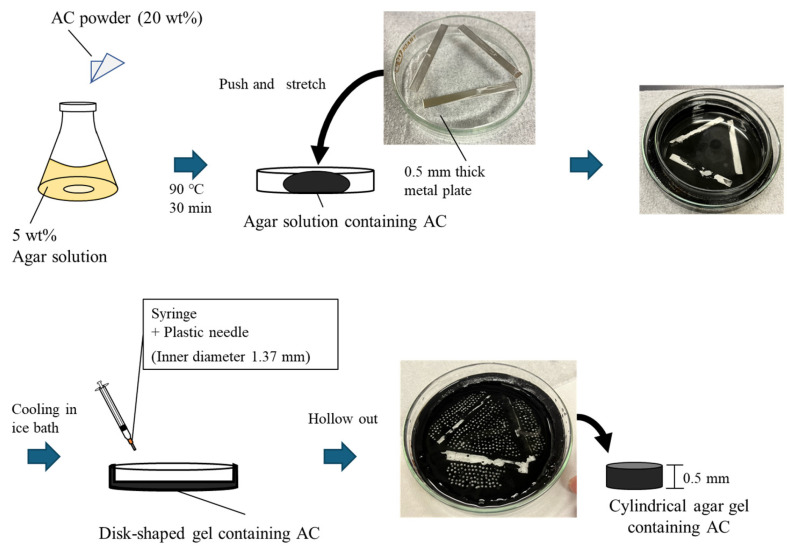
Preparation method of cylindrical agar gel containing AC.

**Figure 6 gels-10-00723-f006:**
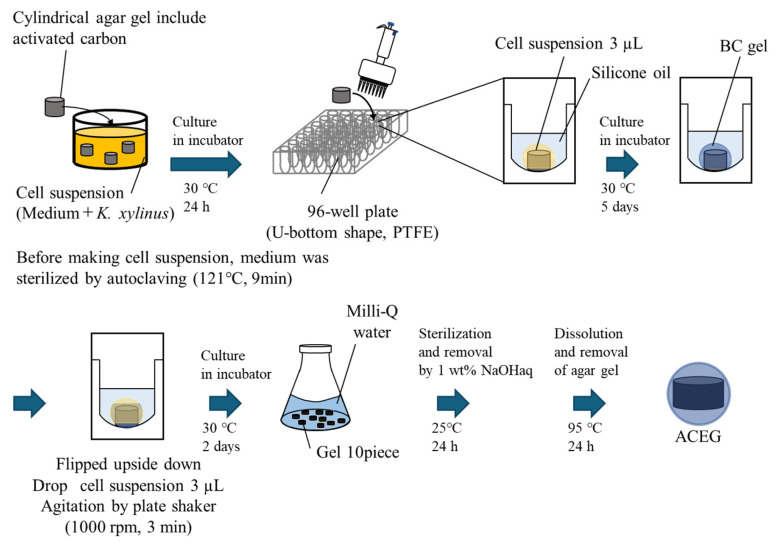
Schematic of Method A for the production of an ACEG.

**Figure 7 gels-10-00723-f007:**
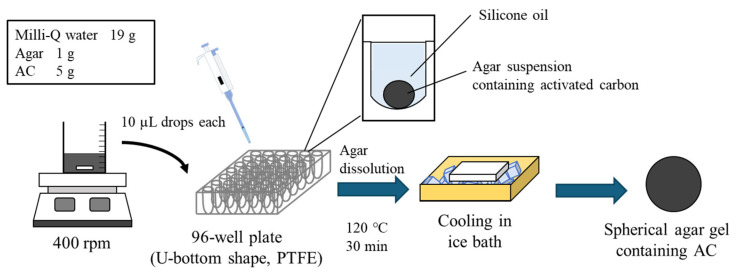
Preparation method of spherical agar gel containing AC.

**Figure 8 gels-10-00723-f008:**
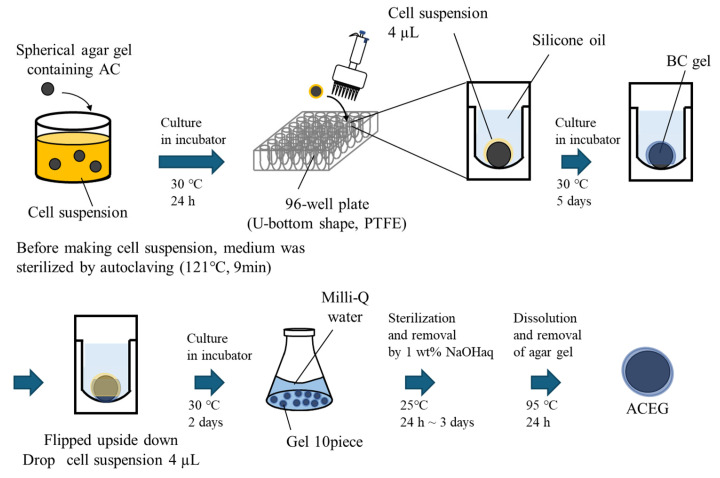
Schematic of method B for the production of an ACEG.

**Table 1 gels-10-00723-t001:** Comparison of AC content in three types of ACEG.

Samples	Activated Carbon Content [µg/piece]	ACEG Volume [mm^3^]	Inclusions Per Unit Volume [µg/mm^3^]
Previously reported gel [22]	289	8.18	35.3
ACEG prepared by Method A	197	3.02	65.1
ACEG prepared by Method B	2140	27.7	77.4

**Table 2 gels-10-00723-t002:** *Q_m_* and *R*^2^ of each ACEG.

Adsorbent	*Q_m_* [mg/mg]	*R* ^2^
Previously reported ACEG	0.378	0.990
ACEG prepared by Method A	0.451	0.956
ACEG prepared by Method B	0.441	0.964

**Table 3 gels-10-00723-t003:** *Q_m_* and *R*^2^ of each AC powder.

Adsorbent	*Q_m_* [mg/mg]	*R* ^2^
Blank AC powder	0.468	0.997
After NaOHaq treatment	0.487	0.999

## Data Availability

The original contributions presented in the study are included in the article, further inquiries can be directed to the corresponding author.

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
