# Peer review of "Improvement of Adsorption Capacity by Refined Encapsulating Method of Activated Carbon into the Hollow-Type Spherical Bacterial Cellulose Gels for Oral Absorbent"

_gels, 2024, doi:10.3390/gels10110723_

Round 1

Reviewer 1 Report

Comments and Suggestions for Authors

In this study, to reduce number of dosing for medications like (KREMEZIN®), as adsorbents to remove uremic toxins in CKD patients, authors have constructed the active carbon-encapsulated hollow spherical bacterial cellulose gel (ACEG). They have tried to increase the amount of adsorbent contained therein, measuring the amount of indole adsorbed by the gel. Please address the questions:

1. Please more explain about bacterial cellulose gels in the abstract. Include some studies in the literature.

2. Authors have not mention the polymers like alginate and its crosslinking potential in the introduction and abstract.

3. Extensive English editing is required. For example, the sentence is a combination of passive and subjective sentences in the abstract: "preparation method of ACEG indole saturation adsorption capacity of the resulting gels was also measured ".

4. What does " NaOHaq " means? Mention the abbreviation. The terms of "nucleous" and "core" is confusing in the abstract.

5. There is no continuity among some paragraphs and the written text in some cases is not scientific, so the revision is needed.

6. Please validate the indole adsorption test and mention the lower and upper range of adsorption.

7. The swelling ratio and degradation of the gel are required.

8. The Rheology characterization is needed.

9. Compare the results with similar studies.

Best

Comments on the Quality of English Language

Extensive English editing is required.

Author Response

Reviewer Report 1

We would like to thank you for your review and answer to each question and comment.

Q1. Please more explain about bacterial cellulose gels in the abstract. Include some studies in the literature.

A1. According to the reviewer’s suggestion, sentence was added to line 15~17.

Q2. Authors have not mention the polymers like alginate and its crosslinking potential in the introduction and abstract.

A2. Thank you for your kind suggestion. We added the sentence to a both introduction and abstract in line17~20 and 84~86 to make it clear.

Q3. Extensive English editing is required. For example, the sentence is a combination of passive and subjective sentences in the abstract: "preparation method of ACEG indole saturation adsorption capacity of the resulting gels was also measured ".

A3. I'm sorry, I have corrected the sentence that you pointed out (line 17-18) and also we proofread the entire text and asked English editing.

Q4. What does " NaOHaq " means? Mention the abbreviation. The terms of "nucleus" and "core" is confusing in the abstract.

A4. I'm sorry for the omission.I fixed it with sodium hydroxide aqueous solution (NaOHaq).

Surely, core and nucleus are coexisting in the text and we unify only “nucleus”.

Q5. There is no continuity among some paragraphs and the written text in some cases is not scientific, so the revision is needed.

A5. We are sorry, it is not easy to read. We revised and inserted some sentences.

Q6. Please validate the indole adsorption test and mention the lower and upper range of adsorption.

A6. According to the reviewer’s indication, the following sentences were added to the line 388~393.

Measurements were performed at various concentrations until adsorption reached saturation. Adsorption isotherms were then plotted, and adsorption was deemed to have reached saturation when the line became flat. In the three conditions, the following indole concentration were used; Method A, 5~50 µg/mL; Method B, 100~1000 µg/mL; and AC powder 250~1000 µg/mL. Compared to method B, method A contained less AC per gel, which is thought to have led to saturation at a lower concentration range.

Q7. The swelling ratio and degradation of the gel are required.

A7. In this experiment, we prepared a bacterial cellulose gel containing activated carbon by devising the culture method. Through experiments, bacterial cellulose gel was kept in the water environment. It is a preliminary experiment, but once dried, bacterial cellulose gel is difficult to completely restore the swelling state. Therefore, we create and save it in an aqueous environment. Therefore, it is not meaningful to measure the swelling degree. ACEG, which adsorbs urine toxins, is finally discharged from the stool. Bacteria cellulose gel is not disassembled within the time of passing the digestive tract. From the above, there was no swelling degree measurement or degradation test.

Q8. The Rheology characterization is needed.

A8. As reviewer’s suggestion, it is interesting to investigate rheology of bacterial cellulose gel. In this experiment, we intended to study on the improvement of encapsulating AC to HSBC. About basic character of bacterial cellulose gel, we are going to investigate in separate experiment.

Q9. Compare the results with similar studies.

A9. As we reported comparison with other absorbent in our previous paper, Wang, W. et al. showed that porous activated carbon effectively removed uremic toxins such as creatinine and uric acid, and, using Langmuir isotherms, they investigated that the level of saturated adsorption of creatinine was 3.81 µmol/mg [1]. This is comparable to our data on indole. Recently, Guangle, Q. et al. summarized the adsorption capacity of uremic toxins in comparison with other types of absorbents [2], and the levels found in this study were higher than those found for the absorbents.

[1]. Wang, W.; Wang, Z.; Li, K.; Liu, Y.; Xie, D.; Shan, S.; He, L.; Mei, Y. Adsorption of uremic toxins using biochar for dialysate regeneration. Biomass Convers. Biorefinery 2023, 13, 11499–11511.

[2].  Guangle, Q.; Gan, Z.; Dapeng, C.; Jingjie, S. Adsorption of uremic toxins by modified activated carbon of different mesh with sulfuric acid. Adsorption 2024.

Reviewer 2 Report

Comments and Suggestions for Authors

The manuscript (gels-3289381) presents an interesting study. However, there are several points that should be addressed to enhance the overall quality, clarity, and readability of the manuscript before it can be reconsidered for publication.

1. Line numbers: Please add line numbers to the manuscript. This will make it easier to provide feedback on specific sections.

2. English language and clarity: The manuscript should undergo a thorough English language review to improve readability and academic tone. In several sections, it’s difficult to fully understand the authors’ intended message. Below are a few examples where language improvements are necessary:

- Page 1 (Introduction, first paragraph): Introduction, first paragraph (Page 1): The sentence “In recent years, Chronic Kidney Disease (CKD) is due to the rapid increase in dialysis patients [1] and the cardiovascular risks involved [2], there is active discussion about measures to address this issue.” should be rephrased for clarity.

- The phrase “There is concern about the effect of immersion in NaOHaq. on the inner AC” should be revised. Also, please specify what specific concerns are being referred to here. Are these related to material degradation, structural integrity, or another aspect?

3. Section 2.1: The sentence “In the case of the previous adjustment method using calcium ions and alginic acid (Ca-Alg gel), a suspension of sodium alginate and AC must be dropped” is unclear. Please clarify the meaning of this phrase.

4. Section 2.1: The discussion about why viscosity increases with the addition of AC should be further discussed. Additionally, please specify the maximum amount of AC that can be effectively loaded or encapsulated within the gel matrix.

5. Section 2.1 (Page 3 and Figure 1): It would be helpful to state the key differences between Method A and Method B to help readers better understand the difference between these two methods. Since these methods are described later in the manuscript, a brief explanation here would improve clarity.

6. Section 4.2: Please specify the amount of AC powder added to the 5 wt% agar solution, as this is important for reproducibility.

7. Section 4.3.2: The statement, “The maximum absorption wavelength (λmax 270 nm) of the solution before and after adsorption was measured using a UV-Vis spectrometer” should be revised. Note that the UV-Vis spectrometer measures absorbance, not wavelength. Additionally, please clarify the way that authors used to quantify concentration. If a calibration curve was used, please provide the equation for reference.

Comments on the Quality of English Language

Please see the 'Comments and Suggestions for Authors' section.

Author Response

We would like to thank you for your review and answer to each question and comment.

Q1. Line numbers: Please add line numbers to the manuscript. This will make it easier to provide feedback on specific sections.

A1. We are sorry for the inconvenience. We submitted our manuscript with line numbers and unexpectedly, they were deleted by unknown accident. If they are deleted again, please ask the editors.

Q2. English language and clarity: The manuscript should undergo a thorough English language review to improve readability and academic tone. In several sections, it’s difficult to fully understand the authors’ intended message. Below are a few examples where language improvements are necessary:

- Page 1 (Introduction, first paragraph): Introduction, first paragraph (Page 1): The sentence “In recent years, Chronic Kidney Disease (CKD) is due to the rapid increase in dialysis patients [1] and the cardiovascular risks involved [2], there is active discussion about measures to address this issue.” should be rephrased for clarity.

- The phrase “There is concern about the effect of immersion in NaOHaq. on the inner AC” should be revised. Also, please specify what specific concerns are being referred to here. Are these related to material degradation, structural integrity, or another aspect?

A2. We are sorry for unclear description and rewrote it in line 29~32 and line 433~435, as you pointed out and also we proofread the entire text and asked English editing.

Q3. Section 2.1: The sentence “In the case of the previous adjustment method using calcium ions and alginic acid (Ca-Alg gel), a suspension of sodium alginate and AC must be dropped” is unclear. Please clarify the meaning of this phrase.

A3. We rewrote it as shown in line 103~106.

Q4. Section 2.1: The discussion about why viscosity increases with the addition of AC should be further discussed. Additionally, please specify the maximum amount of AC that can be effectively loaded or encapsulated within the gel matrix.

A4. Thank you for your important indication. We discussed the reason why the content of AC was limited in nucleus formation. We added the sentences in line148~151, line 174~179.

In Method A, as the amount of AC was increased, the particles came into contact with each other more. This caused the water to lose its fluidity, become more viscous, and turn muddy. In this state, it could no longer be formed into a sheet and could not be hollowed out.

In Method B, the amount of AC that can be encapsulated is a maximum of 20 wt%, the same as in method A. If the concentration is increased beyond this, contact between the cell suspension and the nucleus becomes incomplete, and we have confirmed that encapsulation by the Komagataeibacter xylinus does not proceed well.

Q5. Section 2.1 (Page 3 and Figure 1): It would be helpful to state the key differences between Method A and Method B to help readers better understand the difference between these two methods. Since these methods are described later in the manuscript, a brief explanation here would improve clarity.

A5. According to the reviewer’s suggestion, we added the sentence in line 110~113 to make it clear.

Q6. Section 4.2: Please specify the amount of AC powder added to the 5 wt% agar solution, as this is important for reproducibility.

A6. We are sorry for the lack of detail. and added "20wt% AC powder" in figure 5.

Q7. Section 4.3.2: The statement, “The maximum absorption wavelength (λmax 270 nm) of the solution before and after adsorption was measured using a UV-Vis spectrometer” should be revised. Note that the UV-Vis spectrometer measures absorbance, not wavelength. Additionally, please clarify the way that authors used to quantify concentration. If a calibration curve was used, please provide the equation for reference.

A7. Thank you for your kind suggestion. We rewrote the sentence as shown in line 390~397.

As for the concentration of the indole solution, a calibration curve was created by plotting the relationship between absorbance and indole concentration on a graph, as shown in the text. A new calibration curve equation was created for each measurement, so the same equation was not used. As an example, the calibration curve used in Method B was y=0.0045x+0.013, where y=absorbance [-] and x=indole solution concentration [µg/mL].

Reviewer 3 Report

Comments and Suggestions for Authors

General comments:

- The authors did not report their methods in the text to those existing in the literature. The interpretation of the results was not carried out in the existing scientific context on this topic. For example subchapter 2.1, no references are reported that compare the resulting capsules with those existing in the literature, even if the authors specify the existence of other capsule models.;

- chapter 2.1. it is just a more detailed presentation of the method of obtaining the capsules. There are no pictures taken in the section, there is no SEM, there is no FTIR, and no other method of characterizing the granules obtained, which would certify that they really contain what the authors claim;

- no stability studies of these capsules were done; the authors do not specify whether in the adsorption studies the active carbon remained encapsulated or not; since the authors want to give a medical destination to these capsules, minimal digestibility studies had to be done, to demonstrate to what level of the digestive tract they can reach without being destroyed.

- adsorption studies do not have a reference (a control); the authors do not specify the adsorption capacity of biocellulose. No desorption studies were done to certify saturation capacities, equilibrium, etc.

- no studies were done on the amount of NaOH remaining in the BC fibrils, because the authors did not specify how the NaOH was washed and inactivated from the obtained capsules that were later used in the adsorption studies.

- in chapter 4 to 4.1 the purity of the reagents is not specified; in 4.2 it is not specified anywhere (at any of the stages in which bacterial support was used), what is its concentration.

There is a discrepancy between the text and the numbering of the figures (eg fig 5, in the title of the figure is 6, and the numbering continues from here).

It is specified at one point in the description of method B that a medium is autoclaved, but it is not clear which medium.

Is the successive addition, at intervals of a few days, of bacterial suspension done with fresh suspension?

Before adsorption studies, the capsules are heated to 70 degrees. Why?

There are no bibliographic references for the equations selected to characterize the adsorption process.

As a whole, the article does not clearly present either the capsule production methods or the conditions in which the adsorption experiments were carried out. Key experiments are missing in supporting the information regarding the granule structure and the adsorption process. Correlations with specialized literature are missing.

Comments on the Quality of English Language

Acceptable

Author Response

We would like to thank you for your review and answer to each question and comment.

Q1. The authors did not report their methods in the text to those existing in the literature. The interpretation of the results was not carried out in the existing scientific context on this topic. For example subchapter 2.1, no references are reported that compare the resulting capsules with those existing in the literature, even if the authors specify the existence of other capsule models.;

A1. In this manuscript, we would like to claim the improvement of our previous report. That said the background, highlight and validity of the research. We are sorry for omitting to add the scientific background in this, so we added the them ref [19], [20].

And some sentences were inserted in line 74~76.

Q2. chapter 2.1. it is just a more detailed presentation of the method of obtaining the capsules. There are no pictures taken in the section, there is no SEM, there is no FTIR, and no other method of characterizing the granules obtained, which would certify that they really contain what the authors claim;

A2. It is sure. The SEM images and experimental result of FTIR have been in our previous report.

Toru Hoshi, Kazuyoshi Yamazaki, Yuki Sato, Takaya Shida, Takao Aoyagi.

Production of hollow-type spherical bacterial cellulose as a controlled release device by newly designed floating cultivation. Heliyon 4 (2018) e00873.

doi: 10.1016/j.heliyon.2018. e00873

To avoid repeating we omitted such results.

Q3. no stability studies of these capsules were done; the authors do not specify whether in the adsorption studies the active carbon remained encapsulated or not; since the authors want to give a medical destination to these capsules, minimal digestibility studies had to be done, to demonstrate to what level of the digestive tract they can reach without being destroyed.

A3. In the adsorption test, there was no collapse of the gel or leakage of the encapsulated activated carbon.

For ACEG, in Ref [22], there was no collapse or leakage of the internal activated carbon in the first disintegration test liquid (simulated gastric fluid, pH 1.2), second liquid (simulated intestinal fluid, pH 6.8), or aqueous sodium hydroxide solution (pH 12.6). This suggests that the same is true for ACEG prepared using the new method.

In addition, this report focused on the development of a new preparation method and the evaluation of adsorption capacity. As a future study, we would like to conduct a digestibility test as an approach from a more medical perspective and include it in our next presentation.

Q4. Adsorption studies do not have a reference (a control); the authors do not specify the adsorption capacity of biocellulose. No desorption studies were done to certify saturation capacities, equilibrium, etc.

A4. Our previous report has confirmed that bacteria cellulose gel alone does not adsorbed indole, in Ref [32].

Certainly, about adsorption and desorption of uremic toxins are chemically interesting. However, when considering the adsorption of uremic toxins in the body, the adsorbed ACEG is excreted as a stool. Therefore, we did not focus on desorption.

Q5. no studies were done on the amount of NaOH remaining in the BC fibrils, because the authors did not specify how the NaOH was washed and inactivated from the obtained capsules that were later used in the adsorption studies.

A5. ACEG is immersed in NaOH aqueous solution, and is fully washed using Milli-Q water, removes agar gel, then immersed in the Milli-Q water and stored. This operation suggests that NaOH in BC Fibril is completely removed.

Q6. in chapter 4 to 4.1 the purity of the reagents is not specified; in 4.2 it is not specified anywhere (at any of the stages in which bacterial support was used), what is its concentration.

A6. We added the information of reagents into line 306~308.

Q7. There is a discrepancy between the text and the numbering of the figures (eg fig 5, in the title of the figure is 6, and the numbering continues from here).

A7. We are sorry, layouts of the figures,5 and 6 are not easy to understand. We have revised  them.

Q8. It is specified at one point in the description of method B that a medium is autoclaved, but it is not clear which medium.

A8. Sorry for ambiguous description. We added the information into Figure 6 and 8.

We revised it to “Before making cell suspension, medium was sterilization by autoclaving (121℃, 9min) ”.

Q9. Is the successive addition, at intervals of a few days, of bacterial suspension done with fresh suspension?

A9. The addition of cell suspension was done on the 1st day and after 5 days cultured. The suspension was the same one as the initial cell suspension soaked with the gels. After removing the BC gels biosynthesized at the surface of the culturing petri dish, only the medium was used. Such information was inserted into line 335~338, line 365~368.

Q10. Before adsorption studies, the capsules are heated to 70 degrees. Why?

A10. Heating is performed to wash the activated carbon contained within. If hotter water is used for washing, air bubbles will get inside the ACEG, causing the gel to float on the water. This may result in the uremic toxin adsorption experiment not being performed properly. Therefore, washing was performed at 70 degrees.

Q11. There are no bibliographic references for the equations selected to characterize the adsorption process.

A11. According to other publications, different kinds of absorbents have been used to investigate the adsorption of uremic toxins, as measured using Langmuir isotherms, and they were successful in estimating saturated adsorption. We add the sentence and references [33~37] to make it clear.

Q12. As a whole, the article does not clearly present either the capsule production methods or the conditions in which the adsorption experiments were carried out. Key experiments are missing in supporting the information regarding the granule structure and the adsorption process. Correlations with specialized literature are missing.

A12. We have been researching encapsulation technology using bacterial cellulose gel for over 10 years. At first, we conducted numerous trials and errors to find the best way to create hollow bacterial cellulose gel. We have already published the following reports.

Toru Hoshi, Kazuyoshi Yamazaki, Yuki Sato, Takaya Shida, Takao Aoyagi.

Production of hollow-type spherical bacterial cellulose as a controlled release device by newly designed floating cultivation. Heliyon 4 (2018) e00873.

doi: 10.1016/j.heliyon.2018. e00873

Toru Hoshi, Masashige Suzuki, Mayu Ishikawa, Masahito Endo, Takao Aoyagi.

Encapsulation of Micro-and Milli-Sized Particles with a Hollow-Type Spherical Bacterial Cellulose Gel via Particle-Preloaded Droplet Cultivation. Int. J. Mol. Sci. 2019, 20(19), 4919;

doi: 10.3390/ijms20194919

Toru Hoshi, Masahito Endo, Aya Hirai, Masashige Suzuki, Takao Aoyagi.

Encapsulation of Activated Carbon into a Hollow-Type Spherical Bacterial Cellulose Gel and Its Indole-Adsorption Ability Aimed at Kidney Failure Treatment. Pharmaceutics. 2020 Nov 11;12(11):1076.

doi: 10.3390/pharmaceutics12111076

Aya Hirai, Masashige Suzuki, Kaito Sato, Toru Hoshi, Takao Aoyagi.

Adsorption Capacity of Activated Carbon-Encapsulated Hollow-Type Spherical Bacterial Cellulose Gels for Uremic Toxins in a Simulated Human Gastrointestinal Environment

Gels 2024, 10(7), 417

doi:10.3390/gels10070417

Toru Hoshi, Masashige Suzuki, Takao Aoyagi.

Encapsulation of HRP-Immobilized Silica Particles into Hollow-Type Spherical Bacterial Cellulose Gel: A Novel Approach for Enzyme Reactions within Cellulose Gel Capsules

Gels 2024, 10(8), 516

doi: 10.3390/gels10080516

We have made great efforts to be the first in the world to prepare hollow-type spherical bacterial cellulose gel and have succeeded in doing so by devising innovative cultivation methods. This technology is unique in the world, and can be applied not only to activated carbon, but also to other adsorbents used in medicine. This technology can be applied not only to medicine, but also to food and other fields. In this study, we focused on activated carbon, which is used to treat chronic kidney disease, and investigated effective methods of encapsulation. We are confident that if you follow our series of research, you will understand the importance of this technology.

Round 2

Reviewer 1 Report

Comments and Suggestions for Authors

.

Comments on the Quality of English Language

Minor English editing is needed.

Reviewer 2 Report

Comments and Suggestions for Authors

The authors have provided comprehensive responses to each of the reviewer's comments. Also, I believe now all the concerns raised in the review process have been adequately addressed. So, I recommend accepting the manuscript in its present form.